# Cannabis for the Treatment of Fibromyalgia: A Systematic Review

**DOI:** 10.3390/biomedicines11061621

**Published:** 2023-06-02

**Authors:** Natalie H. Strand, Jillian Maloney, Molly Kraus, Christopher Wie, Michal Turkiewicz, Diego A. Gomez, Olufunmilola Adeleye, Monica W. Harbell

**Affiliations:** 1Department of Anesthesiology and Perioperative Medicine, Mayo Clinic, Phoenix, AZ 85050, USA; 2Mayo Clinic Alix School of Medicine, Phoenix, AZ 85050, USA

**Keywords:** fibromyalgia, cannabis, chronic pain

## Abstract

Fibromyalgia is a common disease syndrome characterized by chronic pain and fatigue in conjunction with cognitive dysfunction such as memory difficulties. Patients currently face a difficult prognosis with limited treatment options and a diminished quality of life. Given its widespread use and potential efficacy in treating other types of pain, cannabis may prove to be an effective treatment for fibromyalgia. This review aims to examine and discuss current clinical evidence regarding the use of cannabis for the treatment of fibromyalgia. An electronic search was conducted on MEDLINE, EMBASE, Cochrane Central Register of Controlled Trials, Cochrane Database of Systematic Reviews, and Scopus using Medical Subject Heading (MeSH) terms on all literature published up to October 2022. A follow-up manual search included a complete verification of relevant studies. The results of four randomized controlled trials (RCTs) and five observational studies (a total of 564 patients) that investigated the effects of cannabis on fibromyalgia symptoms were included in this review. Of the RCTs, only one demonstrated that cannabinoids did not have a different effect than placebo on pain responses. Overall, this analysis shows low-quality evidence supporting short-term pain reduction in people with fibromyalgia treated with cannabinoid therapeutics. Although current evidence is limited, medical cannabis appears to be a safe alternative for treating fibromyalgia.

## 1. Introduction

Fibromyalgia is a common and often debilitating disease syndrome characterized by chronic pain in conjunction with cognitive symptoms including fatigue, impaired sleep, and psychiatric comorbidities [1]. It is estimated to affect 2–8% of the population, with women more commonly affected than men [1]. Fibromyalgia is a clinical diagnosis based on defined criteria including widespread pain, chronicity of symptoms, and lack of an alternative explanation for symptoms [2]. While the pathophysiology of fibromyalgia remains unclear, the pain is thought to be due to derangements in central-nervous-system-mediated pain processing [3]. This disease adversely affects patients’ quality of life and can carry a significant monetary burden [4]. Current standard treatments include cognitive behavioral therapy (CBT), antidepressant and neuropathic medications, and exercise therapy. However, the efficacy of these treatments is limited [5]. Patients currently face a difficult prognosis with limited treatment options and often diminished quality of life.

Cannabis has been proposed as an alternative therapy for the treatment of fibromyalgia. While the cannabis plant has a complex chemical makeup, the two components most commonly isolated for therapeutic effects are delta-9-tetrahydrocannabinol (THC) and cannabidiol (CBD) [6]. Their effects are mediated through the type 1 cannabinoid receptor (CB1), which is expressed in the central nervous system, and the type 2 cannabinoid receptor (CB2), which is expressed in peripheral inflammatory cells [7].

The use of medical cannabis has been explored in a wide variety of diseases. Currently, THC has the Food and Drug Administration (FDA)’s approval for the treatment of chemotherapy-induced nausea and vomiting, and for appetite stimulation related to cancer, although it is not considered a first-line treatment [8]. It has also been explored in a variety of chronic pain disorders, such as muscle spasticity from multiple sclerosis [9] and neuropathic pain [10]. Given its widespread use and potential efficacy in treating other types of pain, there is speculation that cannabis may prove to be an effective treatment for fibromyalgia. The aim of this review is to examine current clinical evidence regarding the use of cannabis for the treatment of fibromyalgia.

## 2. Methods

A systematic search was conducted using the Preferred Reporting Items for Systematic Reviews and Meta-Analyses (PRISMA) 2020 guidelines to identify relevant literature. The databases searched were MEDLINE, EMBASE, Cochrane Central Register of Controlled Trials, Cochrane Database of Systematic Reviews, and Scopus. The review focused on peer-reviewed studies published in the English language from database inception until 25 September 2022. The search utilized the following keywords: (fibromyalgia) OR (central sensitization) OR (chronic pain) AND (medical cannabis) OR (cannabis) OR (cannabinoids) AND (randomized controlled trial) OR (controlled clinical trial) OR (randomized) OR (placebo) OR (randomly) OR (trial). Two reviewers independently extracted the data, and any discrepancies were resolved by a third review author. The authors conducted the latest database search on 23 October 2022, before submission, and the details of the search strategy are presented in Appendix A.

Original articles were included if they met the following criteria: (1) observational, retrospective, or prospective human study design; (2) published in the English language; (3) the target population comprised patients diagnosed with fibromyalgia; and (4) the study assessed the effects of cannabinoid products on fibromyalgia pain. The primary outcomes of interest included widespread pain index (WPI), symptom severity scale (SS), Fibromyalgia Impact Questionnaire (FIQ) score, and visual analog scale (VAS) pain. Exclusion criteria included duplicated studies, narrative reviews, integrative reviews, letters, systematic reviews, and meta-analyses, as well as animal and cell studies.

Two authors (D.A.G. and V.T.F.) independently reviewed the titles and abstracts of all identified studies for eligibility and, subsequently, assessed each study to determine whether it met the inclusion criteria. The following information was extracted for each study: (1) author, year of publication; (2) sample size, number of patients in FM and control groups, gender, age; (3) diagnostic criteria used to diagnose FM; and (4) data regarding outcomes of interest in both groups.

The risk of bias for this study was evaluated using either the Newcastle–Ottawa Quality Assessment Scale (NOS) or the Cochrane Risk-of-Bias Tool (C-ROB). The NOS was used to assess observational studies, while the C-ROB was used to evaluate randomized controlled trials (RCTs). The NOS evaluation focused on three domains: selection, comparability, and exposure/outcome. The maximum numbers of stars awarded for each domain were four, two, and three, respectively. A higher number of stars indicated a lower risk of bias. The C-ROB evaluated six domains: selection, performance, detection, attrition, reporting, and other biases, each of which could receive a score of high, low, or unclear risk. Two authors completed the assessments independently, with a third author resolving any discrepancies.

To assess the overall quality of evidence for cannabinoids in treating pain in fibromyalgia, the authors employed the Grading of Recommendations Assessment, Development, and Evaluation (GRADE) approach. GRADE evaluates the certainty of evidence using standard criteria and assigns a rating of very low, low, moderate, or high to each outcome.

Table 1 summarizes the risk-of-bias assessment for cohort studies. An adequate follow-up period was defined as at least six months, and a minimum of 95% of participants remaining under observation at the primary endpoint was considered sufficient (i.e., fewer than 5% of patients dropped out). Only one of the five studies [11] included a non-exposed control; thus, its selection of controls and comparability bias were evaluated by the authors. Four of the five studies did not select a control group and, therefore, were not evaluated for selection of controls or comparability. Two of the studies [12,13] began with an analgesic trial phase, after which the patients were initiated on medical cannabis therapy (MCT). The calculations concerning the duration of follow-up and the percentage of participants lost to follow-up were based on the number of patients who entered the MCT phase of the investigation. Out of the five studies analyzed, three exhibited a low risk of selection bias, whereas two [11,13] demonstrated high selection bias. This was attributed to their selection of a non-exposed cohort from a different source, and to their dependence on written self-reports to confirm exposure. All studies demonstrated moderate-to-high risk of bias pertaining to outcomes, owing to a high percentage of patients lost to follow-up [13,14], insufficient follow-up of cohorts [11,15], or inadequate assessment of outcomes [11]. Figure 1 shows the C-ROB assessment of the four included RCTs. All four studies demonstrated a low risk of bias in all domains. However, two studies demonstrated attrition bias due to attrition rates of 18% [16] and 20% [17].

The evaluation conducted using GRADE determined that the evidence supporting the reduction in pain in fibromyalgia through the use of cannabinoids was generally of low quality. Although all studies incorporated in the analysis were prospective, only four of them were randomized controlled trials (RCTs), thereby diminishing the quality of evidence for this intervention. Stratifying the GRADE quality assessment by various methods of pain evaluation—such as the visual analog scale (VAS) and the Fibromyalgia Impact Questionnaire (FIQ)—demonstrated low-quality evidence for the efficacy of cannabis treatment in reducing pain. A higher-quality RCT was identified, yielding moderate support for the reduction in lower-back pain following cannabis treatment in fibromyalgia. Another RCT was identified demonstrating high-quality evidence that cannabis did not decrease the number of tender points. A summary table with the GRADE assessment is displayed in Table 2.

## 3. Results

After duplicates were removed, we identified 1203 articles. After screening and assessing for eligibility, nine studies involving 564 participants were included in the systematic review [11,12,13,14,15,16,17,18,19]. (Figure 2). For a comprehensive list of the excluded articles, see Appendix A. The exclusion criteria were incomplete clinical trials, duplicated studies, narrative reviews, integrative reviews, letters, systematic reviews, and meta-analyses, as well as animal and cell studies.

Of the nine included studies, four were randomized controlled trials and five were observational studies. Of the four randomized controlled trials, two studies used a parallel design [18,19] and two used a crossover design [16,17]. Of the five observational studies, one used a crossover design [12], one used a cross-sectional design [11], and three used a prospective or retrospective design [13,14,15]. Of the eight longitudinal studies, five were short-term in duration (4 to 10 weeks), while three were intermediate-term studies (one was 16 weeks, two were 24 weeks). The study sizes ranged between 18 and 367. Table 3 summarizes the study characteristics.

The mean age of the participants ranged from 33.4 to 52.9 years. Three studies did not report the sex distribution of their participants; of the six that did, the percentage of females ranged from 73% to 100%. It should be noted that race was not reported in any of the studies.

All but one study [12] required that patients were diagnosed with fibromyalgia according to the most recent American College of Rheumatology diagnostic criteria. One study reported an inclusion criterion requiring a defined pain intensity at baseline (score of 5 or above on a 0–10 scale); this study was also one of only two studies to report a criterion that all patients be female [17]. Six studies required for inclusion that the pain was refractory to previous alternative analgesic therapy, without specifying the type and dosage of said therapies.

All but one study [12] listed exclusion criteria. All eight studies excluded people with major medical and psychiatric diseases. Two studies explicitly excluded people with a sensitivity to cannabinoids [16,18]. One study explicitly excluded people with a history of substance abuse [11].

One study excluded patients with recent cannabis use [17]. One study excluded patients with a history of oral cannabinoids for pain management but did not specify whether other routes of administration or use of cannabinoids for other purposes were allowed [19]. One study permitted patients with recent cannabis use to participate following a 2-week washout period [16].

The treatments used in the studies were oral synthetic THC (nabilone) [19] and cannabis in various forms. Treatments were administered as a pill, oil [16,18], smoke [14], or vapor. The cross-sectional study asked participants to clarify whether they consumed oral or inhaled cannabis but did not further specify the type [11]. Two studies focused on inhaled cannabis, with one using either smoked or vaporized cannabis [12] and the other having participants inhale it through a balloon [17]. One study selected the route of cannabis administration based on patient preferences [15]. Of the four randomized controlled trials, three compared to placebos [17,18,19], while one compared to amitriptyline [16].

### Quality Assessment

There was overall low-quality evidence to support reduced pain in fibromyalgia with cannabinoid treatments. While all studies included in this analysis were prospective in nature, only four were RCTs, which reduced the quality of evidence per GRADE. When assessing by various methods of pain evaluation, such as the visual analog scale (VAS) and the Fibromyalgia Impact Questionnaire (FIQ), there was also low-quality evidence to support a reduction in pain with cannabis treatment.

Two randomized controlled trials found that nabilone is an effective, well-tolerated treatment option for pain reduction in fibromyalgia patients. Another randomized controlled trial and all three observational studies showed that cannabinoids improved quality of life and alleviated pain, respectively, in patients with fibromyalgia. Only one randomized controlled trial demonstrated that cannabinoids did not have a different effect than placebo on pain responses [17]. Table 4 summarizes the effects of the cannabis interventions.

## 4. Discussion

This systematic review found low-quality evidence supporting short-term pain reduction in people with fibromyalgia treated with cannabinoid therapeutics. There may be positive effects on measures of quality of life affected by this syndrome, including sleep quality [14,16], mood [11,14,18,19], libido [14,18], and appetite [14]. These improvements were largely inconsistent across studies.

Both THC and CBD use were explored as treatment options for fibromyalgia. THC has been found in studies to positively affect pain regulation, appetite, and mood [20]. CBD has been shown to have both anti-inflammatory and pain-relieving characteristics [21]. THC acts as a partial agonist on CB1 and CB2 receptors. CBD is a negative allosteric modulator of the CB1 receptor. In theory, when combined, the synergistic effects of both components could be beneficial in using cannabis as an anti-nociceptive agent [21]. However, antagonistic interactions between THC and CBD can also occur due to their varying properties. 

One study explored the intricate relationship between THC and CBD using inhaled cannabis with varying dosages of both THC and CBD, including Bedrocan (22.4 mg THC, <1 mg CBD), Bediol (13.4 mg THC, 17.8 mg CBD), Bedrolite (<1 mg THC, 18.4 mg CBD), and a placebo without THC or CBD. The combination of high doses of both THC and CBD in Bediol was found to decrease spontaneous pain significantly. However, the other varieties with high THC or CBD contents did not have a more significant effect than the placebo on pain responses. Notably, Bedrolite—a cannabis variety with a high CBD content—displayed a lack of pain relief [17]. On the other hand, another study found a significant improvement in symptoms and quality of life of fibromyalgia patients, as measured by the Fibromyalgia Impact Questionnaire (FIQ), using THC-rich, low-CBD-dose cannabis oil (mean dose 4.4 mg THC, 0.08 mg CBD). The authors concluded that cannabis oil offered a low-cost and well-tolerated management option for symptom relief and improved the quality of life for patients with fibromyalgia [18]. Although previous literature has reported that patients experience improvements in their chronic pain from CBD-only treatment [22,23], the findings from our review suggest that, currently, the most promising evidence is for THC as a treatment for fibromyalgia. 

Further studies have been carried out to evaluate outcomes with nabilone—a synthetic analog of THC that is considered to be twice as active as THC [24]. A crossover randomized controlled trial of 31 patients treated for two weeks with each drug comparing nabilone to amitriptyline—a tricyclic antidepressant widely used to treat FM—found nabilone to be superior in improving sleep dysfunction. However, there was no effect on pain or mood [16]. The authors suggested that their findings may have been limited by the relatively short duration of drug exposure. In contrast, another RCT of 40 patients with fibromyalgia, investigating the efficacy of nabilone compared to a placebo in terms of pain reduction and quality of life improvement over a 4-week duration, described significant decreases in pain scores and anxiety [19]. Thus, it may be beneficial to design studies with an increased course of treatment in trials. 

A potential downside associated with increasing the duration of treatment is an increased risk of adverse side effects. Both clinicians and patients must know that cannabinoid therapies may evoke various side effects, which is especially important given that patients with fibromyalgia are more sensitive to medications than the general population [25]. The most common side effects seen in this review included drowsiness [11,18,19], dizziness [11,14,18,19], nausea/vomiting [14], dry mouth [14,16,18,19], drug high [11,17], coughing [17], sore throat [12], tachycardia [11], conjunctival irritation [11,12], hypotension [11], and gastrointestinal symptoms [14]. No severe adverse side effects associated with cannabinoids were reported in the studies. Medical cannabis appears to be a safe alternative for treating fibromyalgia. In addition to beginning with low dosages and titration according to clinical symptoms, we recommend regular monitoring to prevent further risks—especially in the treatment of cannabis-naive patients. 

In recent years, changing attitudes about marijuana have led to the use of cannabis becoming progressively more popular in medical settings, especially for conditions that have exhausted conventional therapies. Furthermore, in the context of the opioid crisis, there has been rising interest in the efficacy of cannabis use for chronic pain conditions such as chronic non-cancer pain [26], neuropathic pain in multiple sclerosis [27], and chemotherapy-induced nausea and vomiting [26,28], among others. Despite this heightened interest, there is still inconclusive and often contradictory evidence in the literature, with limited trials having been undertaken. Nevertheless, positive analgesic benefits reported by patients using cannabis to treat chronic pain are well established [29,30].

This systematic review had some limitations. First, our inclusion and exclusion criteria limited the number of studies examined and may have excluded valuable studies on this topic. Secondly, the heterogeneity of the treatments employed—in terms of chemical formulation, dosage, and route of administration—prevented us from making meaningful comparisons between studies. Furthermore, some studies observed notable attrition rates, had insufficient follow-up, and inadequately assessed clinical outcomes, preventing a high-powered robust investigation. 

In conclusion, there remains a potential role for cannabinoids in the management of fibromyalgia, although current evidence is limited. Nonetheless, more research on this topic is needed to confirm the efficacy of cannabinoids, ascertain the most effective THC–CBD formulation, determine a more standardized assessment for clinical outcomes, and analyze long-term outcomes. 

## 5. Conclusions

The use of cannabis in fibromyalgia treatment is still an area of ongoing study. CBD and THC have been studied for their potential therapeutic benefits in a variety of medical conditions with manifestations of pain and sleep disturbances. These cannabinoids interact with the body’s endocannabinoid system, which plays a role in regulating pain, mood, and other physiological processes, suggesting that they could play a role in managing the cardinal symptoms of fibromyalgia.

There remains a growing interest in the use of cannabinoids as potential treatment options for fibromyalgia. While some studies show promising results, others have been inconclusive. Overall, the effectiveness of these cannabinoids in treating fibromyalgia remains uncertain. Our investigation revealed that they may be effective in reducing pain and improving sleep in fibromyalgia patients, but more studies are needed to strengthen these findings.

The use of cannabinoids for medical purposes is still relatively new, and much is still unknown. To understand the potential benefits, risks, and optimal dosages and formulations, there is more work to be done through clinical trials. Overall, there remains a potential role for cannabinoids in the management of fibromyalgia, despite currently limited evidence. Nonetheless, more research on this topic is needed to confirm the efficacy of cannabinoids, ascertain the most effective THC–CBD formulation, determine a more standardized assessment for clinical outcomes, and analyze long-term outcomes.

## Figures and Tables

**Figure 1 biomedicines-11-01621-f001:**
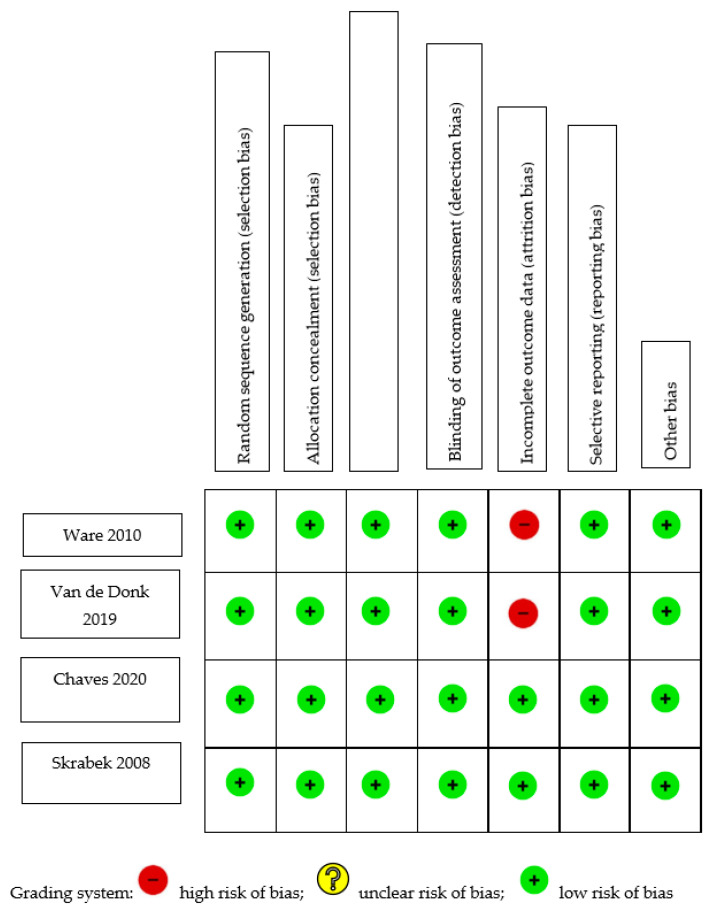
Cochrane Risk-of-Bias Assessment of randomized trials [16,17,18,19].

**Figure 2 biomedicines-11-01621-f002:**
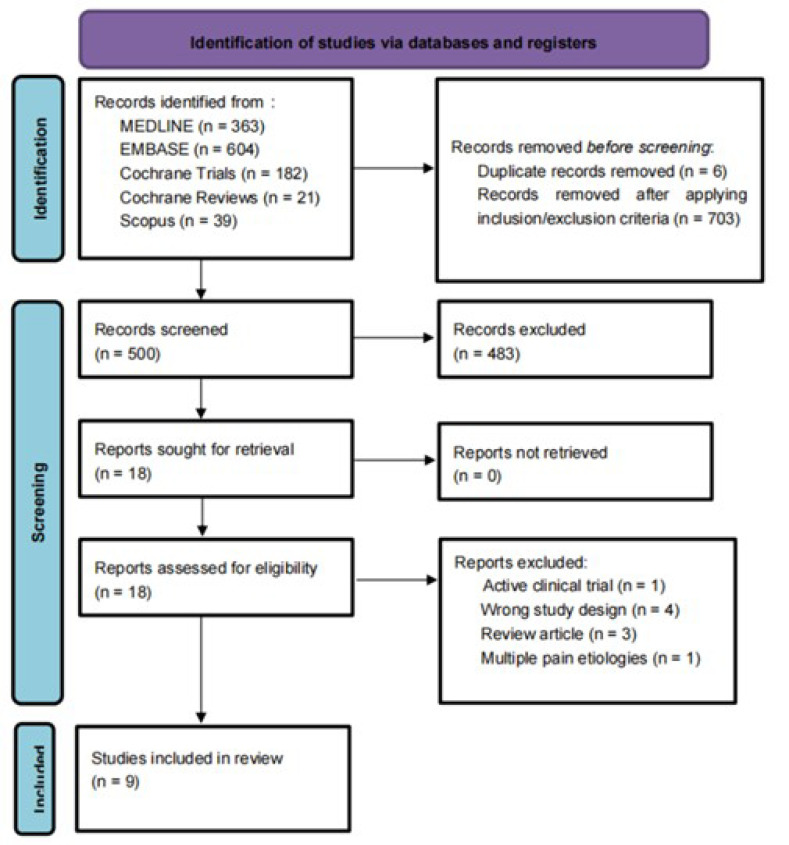
PRISMA flowchart: PRISMA flow diagram for systematic review. Flowchart of the study selection process, inclusion and exclusion of studies, and reasons for exclusion are displayed.

**Table 1 biomedicines-11-01621-t001:** Newcastle–Ottawa risk assessment.

Author	Year	Selection	Comparability	Outcome
Cannabis in Fibromyalgia Studies				
Fiz et al. [11]	2011	*	*	*
Yassin et al. [12]	2019	***	-	**
Habib et al. [13]	2018	*	-	*
Sagy et al. [14]	2019	***	-	**
Hershkovich et al. [15]	2023	***	-	*

Quality of cohort and case–control studies was determined using the Newcastle–Ottawa scale, which evaluates three categories: selection (maximum four stars), comparability (maximum two stars), and outcome (maximum three stars). Stars are denoted as none (-), one (*), two (**), or three (***). Habib: high attrition, follow-up of 10 months. Hershkovich: only 1 month of follow-up.

**Table 2 biomedicines-11-01621-t002:** GRADE assessment.

Certainty Assessment							Impact	Certainty
№ of Studies	Study Design	Risk of Bias	Inconsistency	Indirectness	Imprecision	Other Considerations		
Fibromyalgia Impact Questionnaire (FIQ) Score								
4	2 RCTs and 3 observational studies	Not serious	Serious ^a^	Not serious	Not serious		Three out of the four studies reported reductions in FIQ score. Average score reductions ranged from −12.07 to −45 points.	⨁⨁◯◯ Low
Visual Analog Scale (VAS) for Acute and Chronic Pain								
2	2 RCTs and 2 observational studies	Not serious	Not serious	Not serious	Not serious	Dose–response ^b^	All four studies reported a reduction in pain score ranging from −2.04 to −4.8 cm.	⨁⨁⨁◯ Moderate
Lower-Back Pain								
1	RCT	Not serious	Serious ^c^	Not serious	Not serious	Strong association	Greater decrease in lower-back pain	⨁⨁⨁◯ Moderate
Number of Tender Points								
1	RCT	Not serious	Not serious	Not serious	Not serious	Strong association	No significant decrease in number of tender points	⨁⨁⨁⨁ High

Explanations: a. Two studies employed the Fibromyalgia Impact Questionnaire Revised (FIQR), which places more weight on dysfunction and less on symptoms compared to the previous version of the questionnaire. b. Magnitude of reduction in VAS score was correlated with drug high for Bedrocan and Bediol (van de Donk et al.). c. Lack of standardization of intervention.

**Table 3 biomedicines-11-01621-t003:** Studies and outcomes.

Study Title	Study	Study Design	Diagnostic Criteria	Inclusion Criteria	FM Group	Control Group	# of all Study Patients	Median Age	Gender Distribution	Co-Morbidities	Exclusion Criteria	Intervention	Route of Administration	Follow-Up Length	Concurrent Therapies	Primary Outcome	Secondary Outcomes	Outcomes	Conclusions
Nabilone for the Treatment of Pain in Fibromyalgia	Skrabek et al. (2008) [19]	RCT, parallel	1990 ACR	18–70 years old with FM, pain despite oral medications; no prior use of oral cannabinoids for pain management	Nabilone	Placebo with four-week washout period	40 randomized (20 per group), 33 completed (15 intervention, 18 control)	Mean of study group: 47.6 (9.13); mean of control group: 50.11 (5.96)	NR	NR	Heart disease; schizophrenia or other psychotic disorder; severe liver dysfunction; untreated nonpsychotic emotional disorders; cognitive impairment; major illness in another organ system	0.5 mg nabilone PO at bedtime × 1 week, increased to 0.5 mg BID after 1 week, and uptitration to max dose of 1 mg BID.	Oral	2-week visit, 4-week visit, and 8-week visit after a 4-week washout period	None specified, but patients were allowed to continue using adjunctive therapies for pain treatment during the study	VAS at 2 and 4 weeks: significant decrease at 4 weeks (2.04, *p* < 0.02)	Number of tender points, average tender point pain threshold, Fibromyalgia Impact Questionnaire at 2 and 4 weeks	There were significant decreases in the VAS (2.04, *p* < 0.02), FIQ (12.07, *p* < 0.02) and anxiety (1.67, *p* < 0.02) in the nabilone group at 4 weeks. No significant improvements in the placebo group. More side effects in nabilone group at 2 and 4 weeks (1.58, *p* < 0.02 and 1.54, *p* < 0.05)	Nabilone is a beneficial, tolerable treatment option for fibromyalgia patients, with benefit sin pain relief and functional improvement
The Effects of Nabilone on Sleep in Fibromyalgia: Results of a Randomized Controlled Trial	Ware et al. (2010) [16]	RCT, cross-over	1990 ACR	18+ years old, adult men and non-pregnant women with FM1 and self-reported chronic insomnia. Twenty subjects remained on stable analgesic therapy with negative urine screening for cannabinoids at the baseline visit. Subjects who were using a cannabinoid or amitriptyline at screening underwent a 2-week washout period before entering the study	Nabilone	Amitriptyline 10–20 mg × 2 week with 2-week washout period	32 randomized, 29 completed	49.5 (11.2)	26 F, 5 M, 1 missing data	NR	Cancer pain, unstable cardiac disease, psychotic disorder, schizophrenia, recent manic episode (within the past year), seizure disorder, glaucoma, urinary retention, hypersensitivity to cannabinoids, amitriptyline, or related tricyclic antidepressants, or were taking monoamine oxidase inhibitors	Crossover study comparing nabilone 0.5–1.0 mg for 2 weeks with 2-week washout period with amitriptyline 10–20 mg as an active control. Patients received one of either drug for two weeks, followed by a two-week washout period, and then started the second drug	Oral	10-week study (initial washout period for previous cannabinoid/tricyclic use, two weeks of one drug, intervening washout period, two weeks of second drug, final washout). Data collected every two-week period	20 subjects were on stable analgesic therapy but had to be negative for cannabinoid use at baseline	Quality of sleep (Insomnia Severity Index and Leeds Sleep Evaluation Questionnaire)	Pain, mood, QOL, global satisfaction with treatment, AEs	Nabilone was superior to amitriptyline (Insomnia Severity Index difference 3.2; 95% confidence interval 1.2–5.3). Nabilone was marginally better for restfulness (Leeds Sleep Evaluation Questionnaire difference 0.5 [0.0–1.0]) but not for wakefulness (difference 0.3 [0.2 to 0.8]). No effects on pain, mood, or QOL were observed. AEs were mostly mild–moderate and were more frequent with nabilone. The most common AEs for nabilone were dizziness, nausea, and dry mouth	Nabilone is well tolerated and effective in improving sleep in patients with FM. Low-dose nabilone given once daily at bedtime may be considered as an alternative to amitriptyline. Longer trials are needed to determine the duration of effect and to characterize long-term safety
Cannabis Use in Patients with Fibromyalgia: Effect on Symptoms Relief and Health-Related Quality of Life	Fiz et al. (2011) [11]	OBS, cross-sectional	1990 ACR	18+ years old with moderate–severe FM symptomatology and resistance to pharmacological treatment	Cannabis		56 (28 THC users, 28 non-users)	50 (11.9)	53 females (26 users, 27 non-users) 3 males (2 users, 1 non-user)	NR	Severe illness and history of abuse or dependence for cannabis or other psychoactive substances	Observational study comparing survey responses of cannabis users and nonusers on perceived benefits of cannabis use, adverse effects, and quality of life	Oral (54%), smoked (46%), and combined (43%)		Concurrent therapies used but no significant differences between THC users and non-users	Quality of life (Fibromyalgia Impact Questionnaire and short-form health survey)	Sleep (Pittsburgh sleep quality index) and perceived benefits of cannabis (VAS)	After 2 h of cannabis use, VAS scores showed a statistically significant (*p* = 0.001) reduction in pain and stiffness, enhancement of relaxation, and an increase in somnolence and feeling of wellbeing. Mental health component summary score of the SF-36 was significantly higher (*p* = 0.05) in cannabis users than in non-users. No significant differences were found in the other SF-36 domains, in the FIQ, or in the PSQI	The use of cannabis was associated with beneficial effects on some FM symptoms
An experimental randomized study on the analgesic effects of pharmaceutical-grade cannabis in chronic pain patients with fibromyalgia	Van de Donk et al. (2019) [17]	RCT, cross-over	2010 ACR	18+-year-old females with FM, pain score ≥ 5 for most of the day (pain scale from 0 = no pain to 10 = max)	Each of the subjects tried all 4 substances, including placebo		20	39 (13)	NR	NR	Age < 18 years old, any medical, neurological, or psychiatric illness, use of strong opioids or other painkillers except paracetamol and/or ibuprofen, benzodiazepine use, any known allergies to study medication, illicit drug or alcohol use, recent use of cannabis, pregnancy, breastfeeding, and the presence of pain syndromes other than FM	Patients visited the research unit on 5 occasions. On first visit, the patients were screened and familiarized with the experimental setup. On each of their next visits, the patients received 1 of 4 possible cannabis treatments (in a random order), with at least 2 weeks between visits. These were Bedrocan (22.4 mg THC, <1 mg CBD; Bedrocan International BV, Veendam, the Netherlands), Bediol (13.4 mg THC, 17.8 mg CBD; Bedrocan International BV, Veendam, the Netherlands), Bedrolite (18.4 mg CBD, <1 mg THC; Bedrocan International BV, Veendam, the Netherlands), and a placebo variety without any THC or CBD	Inhaled vapor from balloon	Visits every 2 weeks	Not reported	Relief of experimental pressure pain, electrical pain, and spontaneous pain (VAS)	Plasma concentrations and drug high (Bowdle questionnaire)	No treatment had an effect greater than that of the placebo on spontaneous or electrical pain responses. More subjects receiving Bediol had a 30% decrease in pain scores compared to the placebo (90% vs. 55% of patients, *p* = 0.01), with spontaneous pain scores correlating with the magnitude of drug high (ρ = −0.5, *p* < 0.001). Cannabis treatments containing THC caused a significant increase in pressure pain threshold relative to the placebo (*p* < 0.01). Cannabidiol inhalation increased THC plasma concentrations but diminished THC-induced analgesic effects	None of the treatments were effective in reducing spontaneous pain scores more than the placebo. Further studies are needed to assess efficacy and safety in clinical trials with prolonged treatment periods
Effect of adding medical cannabis to analgesic treatment in patients with low back pain related to fibromyalgia: an observational cross-over single centre study. Clin Exp Rheumatol. 2019 Jan-Feb;37 Suppl 116(1):13–20.	Yassin et al. (2019) [12]	OBS, crossover	Not reported	Symptomatic FM for at least 12 months, symptomatic LBP of more than 12 months between T12 and the gluteal fold, failure of opiate therapy for at least 12 months	N/A	N/A	31	33.4 (12.3)	Not reported	NR	Malignancy, improvement in pain to VAS < 4 after opiate therapy, refusal to undergo medication with duloxetine and opiates, inability to sign an informed consent form, severe cardiovascular disease preventing cannabis administration according to a cardiologist, severe psychiatric disease preventing cannabis administration according to a psychiatrist	3 months of standardized analgesic therapy: 5 mg of oxycodone hydrochloride and 2.5 mg of naloxone hydrochloride twice a day, and duloxetine 30 mg once a day. Following 3 months of this therapy, the patients could opt for medical cannabis (20 g per month) and were treated for a minimum of 6 months	Inhaled (smoking or vaporization)	3- and 6-month follow-up after start of medical cannabis therapy	Oxycodone 5 mg and duloxetine 30 mg	FIQR, VAS, ODI, SF-12, and lumbar range of motion using the modified Schober test	Analgesic drug use was assessed according to the patients’ medical records, including the list of pharmacy-dispensed medications	Medical cannabis therapy allowed a significantly higher improvement in all patient-reported outcomes at 3 months after initiation of medical cannabis, and the improvement was maintained at 6 months. ROM improved after 3 months of medical cannabis therapy and continued to improve at 6 months	Supplementation of analgesic therapy with medical cannabis therapy can alleviate pain in FM patients suffering from LBP
Safety and Efficacy of Medical Cannabis in Fibromyalgia. J Clin Med. 2019 Jun 5;8(6)	Sagy et al. (2019) [14]	OBS	2010 ACR	FM who initiated treatment with medical cannabis from January 2015 to December 2017	N/A	N/A	367	52.9 (8.3)	301 (82%)	Cigarette smokers (37.3%), cancer (9.5%), PTSD (6%)	NR	Low dose of cannabis (e.g., a drop of 15% THC-rich cannabis TID) with gradual increase (e.g., a single drop per day) until they achieved a therapeutic effect (e.g., subjective relief of their pain, significant improvement in their QOL). In case of inflorescence (each cigarette contained 0.75 g of cannabis), patients were instructed to use one breath every 3–4 h, and to increase the amount gradually in this interval until a therapeutic effect was achieved.	Oil, inflorescence (i.e., smoke), or both	6-Month follow-up	Not reported	Treatment response defined as at least moderate or significant improvement in a patient’s condition at 6-month follow-up	Pain intensity—(NRS) Quality of life—global assessment by the patient using the Likert scale, with five options: very good, good, neither good nor bad, bad, or very bad. Perception of the general effect of cannabis—global assessment by using a Likert scale with seven options: significant improvement, moderate improvement, slight improvement, no change, slight deterioration, moderate deterioration, or significant deterioration	Pain intensity (scale 0–10) reduced from a median of 9.0 at baseline to 5.0 (*p* < 0.001), and 194 patients (81.1%) achieved a treatment response. In a multivariate analysis, age above 60 years (odds ratio (OR) 0.34, 95% C.I 0.16–0.72), concerns about cannabis treatment (OR 0.36, 95% C.I 0.16–0.80), spasticity (OR 2.26, 95% C.I 1.08–4.72), and previous use of cannabis (OR 2.46 95% C.I 1.06–5.74) were associated with treatment outcomes. The most common adverse effects were dizziness (7.9%), dry mouth (6.7%), and gastrointestinal symptoms (5.4%)	Medical cannabis appears to be a safe and effective alternative for the treatment of fibromyalgia symptoms
Ingestion of a THC-Rich Cannabis Oil in People with Fibromyalgia: A Randomized, Double-Blind, Placebo-Controlled Clinical Trial. Pain Med. 2020	Chaves et al. (2020) [18]	RCT, parallel	2010 ACR	>18 years old with moderate-to-severe FM symptoms (presenting functional limitation in everyday activities) despite therapies in use, at least one medical or nursing consultation at the health center in the last year	THC-rich cannabis oil	Olive oil	18	51.9 (9.8)	All F	NR	Decompensated organic comorbidities and/or risk of psychiatric conditions (schizophrenia, psychosis, severe personality disorder, current suicidal ideation), another well-defined cause of chronic pain, current pregnancy/lactation, moderate or severe cognitive impairment, and history of cannabinoid sensitivity	The cannabis group received a 30 mL green glass dropper bottle containing cannabis oil (olive oil extraction) of the White Widow variety, at a 24.44 mg/mL concentration of THC and 0.51 mg/mL of CBD—at a proportion of ∼48/1 THC/CBD, along with small quantities of other cannabinoids such as cannabigerol, tetrahydrocannabivarin, cannabinol, and cannabicromen. The initial dose in both groups was one drop (∼1.2 mg of THC and 0.02 mg of CBD) per day sublingually. Participants in both groups were seen at baseline and every 10 days for eight weeks. Dose increases respected the maximum of one drop for each evaluation moment. At each visit, patients filled out the Fibromyalgia Impact Questionnaire (FIQ)	Oral (Oil)	Participants in both groups were seen at baseline and every 10 days for 8 weeks.	Participants in the cannabis group had previously used antidepressants (62.5%), opioids (25%), and benzodiazepines (12.5%). In the placebo group, the rates of the same classes of medications use were 67%, 33%, and 11%, respectively. Patients self-medicated with mild analgesics and anti-inflammatory pills whenever necessary in both groups	Physical function, work status, wellbeing, and associated physical and mental symptoms in FM patients (FIQ)	Clinical and adverse effects	There were no significant differences in baseline FIQ scores between groups. The cannabis group presented a significant decrease in FIQ score in comparison with the placebo group (*p* = 0.005) and in comparison with the cannabis group’s baseline score when compared to post-intervention. (*p* < 0.001). The cannabis group presented significant improvements in the “feel good”, “pain”, “do work”, and “fatigue” scores. The placebo group presented significant improvements in “depression” score after the intervention. There were no severe adverse effects	Phytocannabinoids can be an economically feasible and well-tolerated therapy to reduce symptoms and increase the quality of life of patients with fibromyalgia

RCT, randomized controlled trial; OBS, observational study; NR, not reported; F, female; M, male; FM, fibromyalgia; VAS, visual analog scale; NRS, Numeric Rating Scale; ODI, Oswestry Disability Index; FIQR, Fibromyalgia Impact Questionnaire; THC, Tetrahydrocannabinol; SF-12, Short Form Health Survey; AEs, adverse events; BID, twice a day; PO, per os; QOL, quality of life.

**Table 4 biomedicines-11-01621-t004:** Studies, outcomes, and conclusions.

Study	Study Design	Intervention	Outcomes	Conclusions
Skrabek et al. (2008) [19]	RCT, parallel	Nabilone vs. placebo	Compared to placebo, the nabilone group demonstrated decreases in the VAS (2.04, *p* < 0.02), FIQ (12.07, *p* < 0.02), and anxiety (1.67, *p* < 0.02) at 4 weeks. More side effects were noted in the nabilone group at 2 and 4 weeks, with no major improvements measured in the placebo group	Nabilone is a well-tolerated treatment option for patients with fibromyalgia, with demonstrated benefits in pain relief and functional improvements in quality of life and anxiety
Ware et al. (2010) [16]	RCT, crossover	Nabilone vs. amitriptyline	Though both amitriptyline and nabilone were effective in improving sleep, nabilone proved superior to amitriptyline (Insomnia Severity Index difference 3.2; 95% confidence interval 1.2–5.3). Nabilone was rated better for restfulness (Leeds Sleep Evaluation Questionnaire difference 0.5 [0.0–1.0]), but not for wakefulness (0.3 [0.2 to 0.8]). No effects on pain, mood, or QOL were observed. Side effects were mostly mild–moderate and were more frequent with nabilone, with the most common symptoms being dizziness, nausea, and dry mouth	Nabilone is well tolerated in patients that have fibromyalgia and can improve their sleep. Low-dose nabilone at bedtime may prove an effective alternative to amitriptyline for sleep management in this population
Fiz et al. (2011) [11]	OBS, cross-sectional	Survey of cannabis users vs. non-users	Two hours following cannabis consumption, measured outcomes indicated a statistically significant (*p* = 0.001) improvement in subjects’ reported pain and stiffness, relaxation, and increased somnolence and feeling of wellbeing. Additionally, the mental health component of the SF-36 was significantly higher (*p* = 0.05) in cannabis users than in non-users	Cannabis use was associated with improvements in some fibromyalgia symptoms, including pain, feelings of wellbeing and relaxation, and mental health scores
Van de Donk et al. (2019) [17]	RCT, crossover	Inhaled THC/CBD combo vs. placebo	None of the treatment modalities impacted spontaneous or electrical pain responses when compared to the placebo. Cannabis varieties that contained THC resulted in a significant increase in subjects’ pain threshold for pressure when compared to the placebo (*p* < 0.01). Though inhalation of cannabidiol was noted to increase THC plasma concentrations, the analgesic effects associated with THC were found to be diminished in this route of administration	Analgesic benefits were limited to cannabis varieties containing THC and were observed exclusively in the evoked pressure pain model. None of the different treatments in this study were better than the placebo in improving spontaneous pain scores
Yassin et al. (2019) [12]	OBS, crossover	Inhaled cannabis vs. oxycodone/naloxone/duloxetine	While standard analgesic therapy showed modest improvements in scores when compared to baseline, the addition of medical cannabis resulted in a significantly higher improvement in all patient-reported outcomes at 3 months. This observed improvement was maintained at 6 months. Range of motion improved after 3 months of cannabis therapy and showed continued improvement at 6 months	Adjunct treatment with medical cannabis therapy in combination with other analgesics can alleviate lower-back pain in patients suffering from fibromyalgia
Sagy et al. (2019) [14]	OBS	Cannabis (oil, smoke, or both)	Pain was reduced from a baseline median of 9.0 to 5.0 (*p* < 0.001); 81.1% of patients achieved a treatment response. Factors associated with treatment outcomes included age over 60 years (odds ratio (OR) 0.34, 95% C.I 0.16–0.72), spasticity (OR 2.26, 95% C.I 1.08–4.72), concerns about cannabis treatment (OR 0.36, 95% C.I 0.16–0.80), and previous cannabis use (OR 2.46 95% C.I 1.06–5.74). The most common adverse effects were mild, including dizziness, dry mouth, and GI symptoms	Cannabis was found to be a well-tolerated and effective treatment modality for fibromyalgia symptoms.
Chaves et al. (2020) [18]	RCT, parallel	THC-rich cannabis oil vs. olive oil	There were no significant differences in baseline FIQ score between the groups at the beginning of the study. After the intervention, the cannabis oil group experienced a significant decrease in FIQ score relative to the olive-oil placebo (*p* = 0.005), as well as when compared with their own baseline score (*p* < 0.001). The cannabis group reported significant improvements in the “feel good”, “pain”, “do work”, and “fatigue” scores. The placebo group reported significant improvements in the “depression” score following the intervention. No major adverse effects were observed	Phytocannabinoids can be a low-cost therapy option with minimal adverse effects to alleviate symptoms and increase the quality of life in patients experiencing fibromyalgia

RCT, randomized controlled trial; OBS, observational study; CBD, cannabidiol; THC, tetrahydrocannabinol; VAS, visual analog scale; PSQI, Pittsburgh sleep quality index; FIQR, Fibromyalgia Impact Questionnaire; AEs, adverse events; QOL, quality of life; LBP, lower-back pain; ROM, range of motion.

## Data Availability

Not applicable.

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
