# Peer review of "Cannabis for the Treatment of Fibromyalgia: A Systematic Review"

_biomedicines, 2023, doi:10.3390/biomedicines11061621_

Round 1

Reviewer 1 Report

In this review, the Authors aimed to examine current clinical evidence regarding the use of cannabis for fibromyalgia treatment. 

The topic is interesting and the study is well designed.

Please remove exclusion criteria from results section.

Please provide a flowchart describing review process.

A few tables reporting on included studies populations and their results are mandatory.

Please have the paper checked by an English native speaker to correct grammar and syntactic errors.

Author Response

  1. Please remove exclusion criteria from results section. ** They have been removed.
  2. Please provide a flowchart describing review process. This has been included. "Prisma Flow Diagram"
  3. A few tables reporting on included studies populations and their results are mandatory. Have added figure 2, Table 1, Table 2, Table 3, Table 4, Table S1, and Table S2
  4. This has been completed. 

Reviewer 2 Report

The text is well written.

Some thoughts

The introduction and summary are short and authors should consider expanding them.

The text reports data not available for review. Table 1,2, supplementary table 2, table 3 and 4 were not found. It is not known whether these tables include studies and their eligibility criteria. Even if submitted as supplementary material, they should appear in the text.

The PRISMA flowchart is not included in the study.

Usually retrospective studies cover the last five years. The text is not clear about the time period under consideration.

The registration number in the PROSPERO survey database is not listed. Is the study approved or not?

References 22 and 23 need correction.

Abbreviations at the end of text are not elegant. Abbreviations should be there as soon as the term appears and then used.

Conclusions should be written for the study.

Author Response

  1. The text is well written.** Thank you.
  2. The introduction and summary are short and authors should consider expanding them.** Thank you. This has been addressed. 
  3. The text reports data not available for review. Table 1,2, supplementary table 2, table 3 and 4 were not found. It is not known whether these tables include studies and their eligibility criteria. Even if submitted as supplementary material, they should appear in the text. ** These have been included. Thank you. 
  4. The PRISMA flowchart is not included in the study.**This has been included. Thank you. 
  5. Usually retrospective studies cover the last five years. The text is not clear about the time period under consideration.**We did not include a time limit as there is such a paucity of information regarding this data. Thus, all eligible studies were included regardless of date of publication. 
  6. The registration number in the PROSPERO survey database is not listed. Is the study approved or not?
  7. References 22 and 23 need correction. ** These have been corrected in APA style. 
  8. Abbreviations at the end of text are not elegant. Abbreviations should be there as soon as the term appears and then used.
  9. Conclusions should be written for the study. ** This has been addressed. 

Reviewer 3 Report

Strand et al. presented Cannabis for the Treatment of Fibromyalgia, A Systematic Review. The topic is interesting, and the paper is well written. However, in the submitted manuscript, there are no tables or diagrams or supplemental file. Therefore, I may not complete my review.

I will be happy to review the revised manuscript with all tables and diagrams.

Author Response

However, in the submitted manuscript, there are no tables or diagrams or supplemental file. Therefore, I may not complete my review. ** We apologize for this error. The figures and tables are now included. 

Round 2

Reviewer 2 Report

Most fixes included.